# Adapting Brain-Like Neural Networks for Modeling Cortical Visual Prostheses

**Jacob Granley**
Department of Computer Science
University of California Santa Barbara
jgranley@ucsb.edu

**Alexander Riedel**
Ernst-Abbe-University, Jena, Germany
alexander.riedel@eah-jena.de

**Michael Beyeler**
Department of Computer Science
Department of Psychology and Brain Sciences
University of California Santa Barbara
mbeyeler@ucsb.edu

## Abstract

Cortical prostheses are devices implanted in the visual cortex that attempt to restore lost vision by electrically stimulating neurons. Currently, the vision provided by these devices is limited, and accurately predicting the visual percepts resulting from stimulation is an open challenge. We propose to address this challenge by utilizing 'brain-like' convolutional neural networks (CNNs), which have emerged as promising models of the visual system. To investigate the feasibility of adapting brain-like CNNs for modeling visual prostheses, we developed a proof-of-concept model to predict the perceptions resulting from electrical stimulation. We show that a neurologically-inspired decoding of CNN activations produces qualitatively accurate phosphenes, comparable to phosphenes reported by real patients. Overall, this is an essential first step towards building brain-like models of electrical stimulation, which may not just improve the quality of vision provided by cortical prostheses but could also further our understanding of the neural code of vision.

## 1    Introduction

Visual neuroprostheses are emerging as a promising treatment option for restoring visual function lost to injury or disease. Analogous to cochlear implants, cortical prostheses electrically stimulate neurons in the early visual system, typically in the primary (V1) or secondary visual cortex (V2), to elicit neuronal responses that the brain interprets as visual percepts ('phosphenes') [11]. Several devices are currently in development, such as the Orion Visual Cortical Prosthesis System [1], which is based on relatively large surface electrodes, and CORTIVIS  [12, 13], which is based on the Utah array of intracortical penetrating electrodes.

Despite recent technological advances, the vision restored by these devices remains restricted to white-ish or yellow-ish phosphenes of simple geometric shape [1, 6, 13]. The inability to generate more complex visual patterns may at least be partially due to our limited understanding of how visual prostheses interact with the human visual system to shape perception. Current devices might indiscriminately stimulate diverse populations of cortical neurons, each with their own complex neuronal response properties that are often not fully understood [8]. Constructing accurate phosphene models, which predict the appearance of percepts elicited from electrical stimuli directly from

4th Workshop on Shared Visual Representations in Human and Machine Visual Intelligence (SVRHM) at the Neural Information Processing Systems (NeurIPS) conference 2022. New Orleans.

patient data remains one of the biggest challenges in the field, primarily due to the difficulty of data acquisition, the amount of noise in perceptual measures, and patient-to-patient variability.

Alternatively, convolutional neural networks (CNNs) have emerged as promising models of the visual system, allowing the simulation of the natural single-unit cortical response to ecological visual stimuli. To this end, "Brain-Score" [21, 22] provides a comprehensive benchmark that measures how closely activation of intermediate layers in CNNs trained for object classification corresponds to primate cortical and behavioral responses across multiple datasets. If adapted properly, these neural networks could be powerful tools for understanding the neuronal and perceptual effects of electrical stimulation on the visual cortex.

Here, we investigate the feasibility of adapting brain-like deep neural networks (DNNs) for modeling visual cortical prostheses. We make the following contributions:

- We develop techniques facilitating the use of brain-like DNNs trained on object classification to predict visual responses to electrical stimulation. We highlight several key challenges associated with this process, such as strategies for mapping stimulation intensities and anatomical locations to DNN neurons.

- We implement a proof-of-concept phosphene model and compare its predictions to phosphenes reported by real patients. We show that many of the qualitative features that characterize clinically observed phosphenes naturally emerge from a neurologically-inspired decoding of predicted neuronal responses, such as yellow-white, circular phosphenes, and 'shimmering' or 'fireworks'-like qualities of phosphenes. In addition, our model reproduces observed and expected trends in phosphene size.

- We discuss the unique opportunities that brain-like phosphene models offer, which could have implications both in neuroscience (*i.e.*, illuminating the decoding methods the brain uses to interpret unusual activation patterns) and in machine learning (*i.e.*, benchmarks measuring DNN's ability to reproduce the cortical behaviors observed in prosthesis patients).

## 2 Methods

In this section, we discuss a number of the challenges associated with adapting an object classification DNN to be used for modeling of electrical stimulation. To the best of our knowledge, we are the first to attempt to model cortical prostheses with brain-like DNNs. Therefore, we also describe many potential solution techniques, and motivate the choices used in our phosphene model.

**Cortical Prostheses**    We simulated electrical stimulation with two cortical implants: Orion [1] and CORTIVIS [13]. The Orion implant has 60 large surface electrodes (1 mm diameter) spaced $4.2\,\text{mm}$ apart horizontally, and $3\,\text{mm}$ apart diagonally. CORTIVIS is based on the Utah array, which consists of 96 small penetrating electrodes on a square grid spaced $0.4\,\text{mm}$ apart.

**Choice of Brain-Like Model**    We used the `effnetb1` network [20], which is currently the best performing model on `brain-score.org` [21, 22]. We chose this over other networks that directly predict neural activations due to the correspondence of multiple cortical and artificial layers (V1, V2, V4, and IT) and its thorough evaluation across multiple electrophysiological and behavioral datasets. The network is an EfficientNet B1, consisting of 7.8M parameters, with adversarially-robust training as described in [20]. We focused on the layers corresponding to cortical areas V1 and V2 (`blocks.3.0` and `blocks.3`), since these are the most relevant sites for cortical prostheses.

**Mapping Artificial Neurons to Anatomical Locations**    Mapping electrical stimulation patterns to DNN activations is a primary challenge associated with our approach. One obvious obstacle is that there is no 1:1 correspondence between artificial and biological neurons. In other words, the corresponding anatomical distribution of the artificial neurons across the cortical surface is unknown.

Previous works have successfully used a wiring length constraint to enforce topographic organization of IT neurons [4, 18], but it is unclear how this might extend to other cortical layers. Further, a method that infers anatomical location after training of the DNN is preferable for its compatibility with existing models. One such method would be to effectively 'reverse' the Brain-Score metrics by training a regression model to associate measured cortical responses with artificial neuron activations.

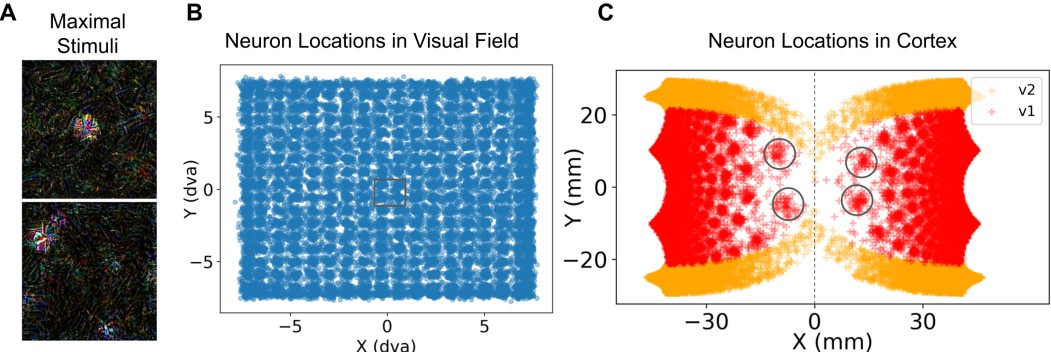

Figure 1: Mapping artificial neurons from a brain-like DNN to anatomical locations in visual cortex. *A*: Examples of stimuli that maximally activate V1 neurons. Each neuron's receptive field (RF) center is defined as the center of mass (marked in red). *B*: Inferred RF centers of all 51,840 V1 and V2 artificial neurons in visual field coordinates (degrees of visual angle; dva). *C*: The corresponding anatomical locations in V1 and V2, under the Wedge-Dipole visuotopic mapping [19], vertical line marking the discontinuous boundary between hemispheres. The 4 clusters (corresponding to channels) of neurons boxed in *B* correspond to the circled clusters in *C*.

While promising, this approach is limited by the specific cortical dataset and regression model used, which might not generalize well in the absence of patient-specific cortical responses.

We therefore propose a new approach suited for the general patient, which can be applied after training of the DNN, and can be made patient-specific without requiring neuronal responses, as illustrated in Fig. 1. In this approach, artificial neurons are mapped to the visual field using an activation maximization technique, and then projected to the cortical surface using well-established visuotopic mappings. First, gradient descent was used to find the stimuli that maximally activate each artificial neuron [24], similar to feature visualization techniques [9]. We used the Adam optimizer with an initial learning rate of 0.1, decreasing by a factor of 5 whenever the loss failed to decrease for 10 iterations. We also enforced L2 regularization on the stimuli, encouraging irrelevant areas in the stimulus to be dark. The vast majority of resulting 'maximal stimuli' were found to be circular and well localized (Fig. 1a). These receptive fields (RFs) were slightly larger than primate V1 and V2 RFs, but with an increase in size (33%) from V1 and V2 that is similar to primate cortex [23]. We also input each stimulus back into the network to verify that it selectively activated only the corresponding artificial neuron. Each stimulus's center of mass was used as the neuron's location in visual field (Fig. 1B). To convert this to a cortical location (Fig. 1C), we used a general Wedge-Dipole mapping [19]—but note that if fMRI data were available, then recent patient-specific mapping techniques (e.g., [2]) would serve as suitable replacements. The RF centers are nearly uniformly spread across the visual field, but are disproportionately sparse in the central foveal cortex region due to cortical magnification.

**Cortical Activation Patterns**   Previous work has demonstrated that cortical electrodes sparsely activate neurons within a certain radius of electrodes, sometimes as far as millimeters away [16]. We therefore modeled cortical activation patterns as Gaussians centered on each stimulating electrode, with standard deviation ($\rho$) as an adjustable parameter, allowing us to account for a range of possible implants. The stimulation intensity $I$ at cortical location $\mathbf{x}$ in mm was given by

$$I(\mathbf{x}; \rho) = \sum_{e \in E} a_e \exp\left(\frac{-||\mathbf{x} - \mathbf{e}||_2^2}{2\rho^2}\right), \tag{1}$$

where $E$ is the set of electrodes, $\mathbf{e}$ denotes the location of each electrode, and $a_e$ is the amplitude of each electrode. We set $\rho$ equal to the electrode spacing of the two cortical prostheses studied (i.e., $4.2\,\mathrm{mm}$ for Orion and $0.4\,\mathrm{mm}$ for CORTIVIS), and otherwise treated stimulation as identical.

**Electrical Stimulation and Phosphene Model**   The final challenge is mapping cortical activation patterns to the artificial neurons and reconstructing the resulting phosphenes. One straightforward

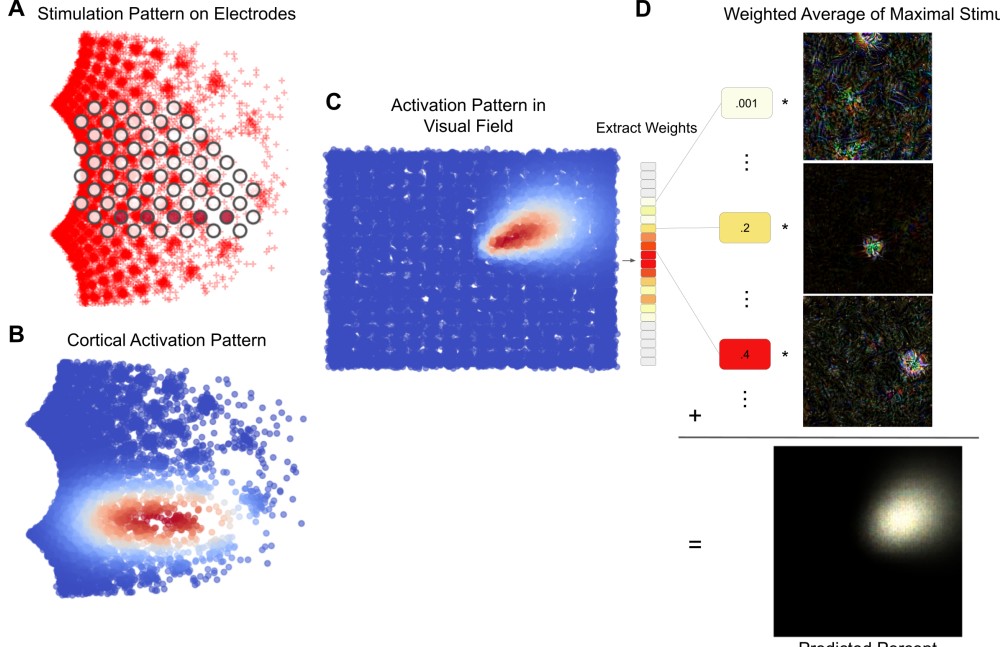

Figure 2: Simulating electrical stimulation of early visual cortex to produce phosphenes. To obtain the predicted phosphene for a electrode activation pattern (**A**), the stimulation intensity at each artificial neuron is computed using Eq. 1. Shown is the stimulation intensity across the neurons in V1 (**B**) and the same neurons in visual field (**C**). The intensities at each neuron are then used as the weight of the corresponding neuron's maximally activating stimulus. The weighted average of all stimuli is output as the final predicted percept (**D**), which spans $16 \times 16$ degrees of visual angle.

solution motivated by previous literature [3, 5, 14] would be to model current spread on the cortical surface, and 'copy and paste' the current intensity onto the corresponding DNN activation maps. However, what it means for an artificial neuron to be 'activated' differs from biological neurons (e.g., artificial neurons can be positively or negatively activated, each neuron is scaled differently), and is dependent on network architecture and each layer's activation function. Further we empirically found that artificial activation patterns from this technique did not produce realistic phosphenes.

We instead propose a neurologically-inspired linear decoding technique that is not architecture-dependent (Fig. 2). To visualize the phosphene ($P$) resulting from stimulation, we average all of the previously computed maximal stimuli ($S$), weighted by the stimulation intensity ($I$) at each neuron's corresponding location in V1 or V2, normalized by activity in a local neighborhood:

$$P_i = \frac{\sum_j I_j * S_{j,i}}{\sum_{j \in \mathcal{N}(i)} I_j}, \tag{2}$$

where the index $i$ iterates over pixels, $j$ iterates over different DNN neurons, and $\mathcal{N}(i)$ denotes all DNN neurons within the local neighborhood of $i$. We found that limiting each maximal stimulus to the 99th percentile of measured RF sizes produced high quality reconstructions.

This decoding strategy is consistent with the idea that each neuron's activation level is indicative of the likelihood that said neuron encountered its preferred stimulus; the resulting percept is thus a weighted sum of preferred stimuli [17]. Normalization by activity within a local neighborhood bears some resemblance to divisive normalization [7]. A benefit of this strategy is its uniform treatment of different cortical areas, allowing for seamless modeling of implants on the border of V1 and V2.

## 3   Results

Fig. 3 shows representative examples of single-electrode phosphenes for Orion and CORTIVIS implanted parafoveally in V1. Prediction for a single phosphene took approximately 3s on CPU. In

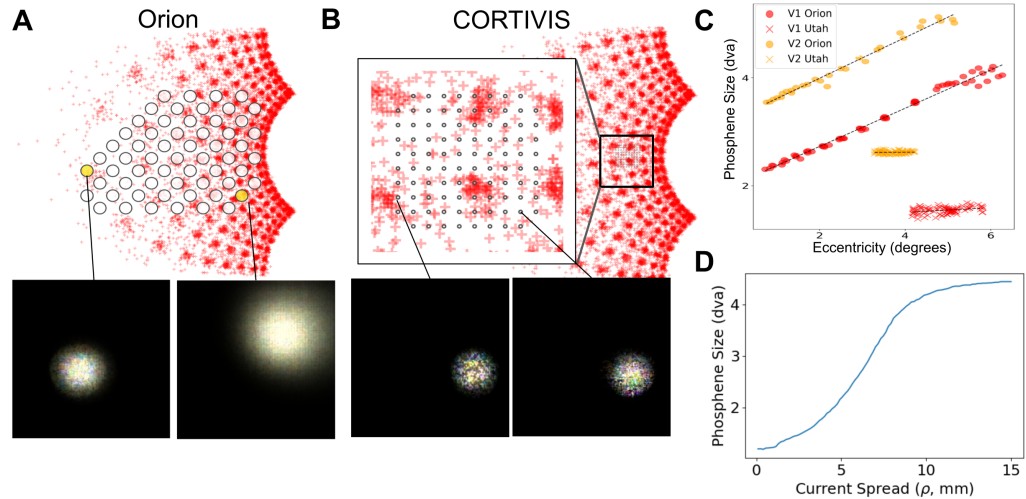

Figure 3: Predicted phosphenes for Orion (**A**) and CORTIVIS (**B**). **C**: Predicted phosphene size in V1 to V2 (implant location not shown) increases with eccentricity. **D**: Predicted phosphene size (measured as radius of the phosphene's convex hull) in V1 as a function of current spread ($\rho$).

general, phosphenes were round yellow-white blobs as expected, but with some subtle high frequency details, such as intermittent patches of bright colors. While predicted Orion phosphenes tended to be highly saturated and large blobs, predicted CORTIVIS phosphenes appeared sparser, smaller, and slightly more colorful. When we simulated the two implants at the same visual field location in V2, phosphenes were qualitatively similar, but 41% larger on average. Due to cortical magnification, electrodes near the fovea may produce smaller phosphenes than electrodes near the periphery. This relationship was linear and strong in Orion for V1 and V2 (t-test, $n = 90$, $p < .001$) matching previous studies [5], but not for the CORTIVIS ($p = 0.803$).

While it is well known that phosphene size saturates with increasing current spread [5], it is less clear whether phosphenes can become arbitrarily small. In theory, the smallest phosphene elicited by stimulating a single neuron should be limited by that neuron's RF size (because the neuron cannot sense which subregion of its RF was stimulated). Consistent with these theoretical considerations, our model predicts phosphene size to saturate both with high and low current spreads (Fig. 3D).

## 4   Discussion

In this work we adapt a brain-like DNN [20] as a decoding model for visual cortical prostheses. We show that our model makes a number of predictions that agree well with the existing neuroscience literature. First, predicted phosphenes are yellow-ish white blobs. Notably, this is not a manually designed feature such as in other models, but arises naturally from the proposed decoding scheme. Second, predicted phosphenes have high-frequency details, such as 'shimmering' or 'fireworks' (patches of bright colors) that qualitatively match many patients' descriptions of observed phosphenes [10]. Third, the model reproduces trends in phosphene size that have either been observed in cortical prostheses [5] or are in line with our theoretical understanding of neuronal population codes.

Our model generally produces similar phosphenes to existing, simpler models (e.g. linear-nonlinear models [6]). However, rather than being hard-coded, these features naturally emerge from our population decoding scheme. Our model's ability to reproduce some of the subtler details of phosphenes inspires hope that this approach might serve as a reliable phosphene model for a range of experiments, enabling improvements in cortical prostheses. For example, if the phosphenes are accurate, then a similar brain-like phosphene model could be used in an end-to-end stimulus optimization framework [15] to discover superior stimulus encoding algorithms.

This study is inherently preliminary as it is limited to a single DNN and has yet to be comprehensively validated against a large dataset of real phosphenes (which currently does not exist). Further, it must

be demonstrated that our results are not just due to peculiarities of the network presented in [20], though our preliminary investigations suggest that other brain-like models make similar predictions.

The intersection of cortical prostheses and DNNs offers a unique set of opportunities for advancing our knowledge of neuroscience. On the one hand, visual prostheses offer a safe way to stimulate and record from visual cortex of awake humans, which may serve as a meaningful benchmark dataset for 'brain-like' DNNs. On the other hand, building brain-like models of electrical stimulation may not only improve the quality of vision provided by cortical prostheses but also offer a unique testbed to further our understanding of the neural code of vision.

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
