# OpenReview forum: "Adapting Brain-Like Neural Networks for Modeling Cortical Visual Prostheses"
_NeurIPS.cc/2022/Workshop/SVRHM — SVRHM Poster_

### Official Review · Reviewer_fzrQ · 2022-10-05
**Novel preliminary research using DNNs to model prostheses**

**Rating:** 7
**Confidence:** 5

**Review:**

Very clearly written and easy to follow. Design choices were justified and the results are promising for a novel application of brain-like DNNs.

**Summary**

This research is a promising start in a creative direction of using brain-like DNNs as models of visual protheses. However, while results trend in the biologically relevant way, it remains challenging to quantify this research with only qualitative observations of generated phosphenes and the trend in size differences between V1 and V2.

**Pros**

Clearly laid out methodology and design choice. Unique and novel approach to leveraging brain-like models. The authors present a very strong prototype for this branch of research.

Results corroborate current literature without having to be hard coded into the model I.e. the color and size trends of the phosphenes

**Cons**

Lack of biological data. Although the authors do declare that this is a major limitation, it is difficult to quantitatively analyze whether this encoding truly does capture the true nature of phosphenes. However, it is promising that the size trends they report are similar observed cortical data.

Did not clarify what set of stimuli was used to activate artificial neurons

---

### Official Review · Reviewer_YAfL · 2022-10-13
**Interesting visual prostheses application of neural network model**

**Rating:** 6
**Confidence:** 3

**Review:**

One of the challenges in cortical visual prostheses is predicting the visual percepts from the stimulation. In this work, the authors adopt a “brain-like’ neural network model to simulate the percept of the stimulation and argued that the phosphenes created by the neural network model is qualitatively similar to the phosphenes perceived by real patients. While this is a very interesting application of computational visual neuroscience, the conclusions drawn should be warranted by further analyzes.
Pros:
•	The Phosphene model suggested in the paper is helpful not only with the application here. If the model does really resemble the perceived phosphenes, this may inform us on how to match the information encoded in the brain and the network. The neurologically inspired linear decoding technique can be the key of the mapping.
•	This method can shed light on some theoretical and technical questions in computational neuroscience. For example, whether the “brain-score” is a good measure of similarity between AI models and the brain. Furthermore, this may inform us how to build better models to resemble neural activities.
•	The paper is clearly written.
Cons:
•	In this paper, the authors argue qualitatively that the phosphenes created by the model are similar to the ones that are perceived by the patients. Although I am not familiar in how to quantitatively evaluate the similarity between the generated phosphenes and the perceived ones from patients, additional experiments and analyzes are needed to secure the conclusion that these models create realistic phosphenes.
•	In the method and result session, the authors mention the simulations of two cortical implants; however, there are no following discussions on that, while the results seem to be different between the different implants.
•	As the authors have already mentioned, it will be helpful to compare the performance with different neural network architecture. This will give us more insight on how the neural networks is relating to the brain.

---

### Official Review · Reviewer_RrBV · 2022-10-14
**Connecting deep network models of visual cortex to visual cortical prostheses**

**Rating:** 7
**Confidence:** 3

**Review:**

### Summary

To build visual cortical prostheses, one must understand the relationship between cortical stimulation and the visual percept. Towards that goal, the authors proposed a novel technique to predict the size and the location of the phosphenes that people might perceive given a particular stimulation pattern in V1 and in V2. They show that phosphenes generated using their technique qualitatively recapitulate known properties of phosphenes reported by patients. For example, they showed that depending on the electrodes used, phosphene size is correlated with distance from foveal representation in early visual cortex. It would be great to see how well these predictions quantitatively match the phosphene shapes and sizes observed in human experiments.

Overall, I thought the paper was clearly written, easy to follow and on a topic very relevant and of interest to the SVRHM audience.

### Some questions (in no particular order)
1. How was the size (in units of degrees of visual angle) of the simulated cortical sheet determined? Data show that the fidelity (in the sense of neural predictivity) of a model of V1 depends on the resolution of the input that is used (cf. Figure 5 in Cadena et al., 2019). How would the assumed size of the cortical sheet affect the data shown in Figure 3, if at all?
2. Some of the high-frequency features in the optimal stimuli may be related to the regularization that is used during stimulus generation. Have you experimented with other techniques and regularization methods for optimal stimulus generation (see, e.g., Bashivan et al., 2019; Olah et al., 2017)? Are the high-frequency features in the optimal stimuli still persistent when other regularization methods are used?
3. What do you think contributed to the lack of correlation between phosphene size and eccentricity when simulations are performed on the Utah array? If you changed the spread of the stimulation for CORTIVIS to be the same as that of Orion (in physical space) instead of setting it to be the distance between electrodes, do you think you would see a correlation between phosphene size and eccentricity? i.e., change $\rho$ in Equation 1 from 0.4 mm to 4.2 mm for CORTIVIS and compute phosphene size as a function of eccentricity.
4. The saturation effect shown by Bosking et al. (2017) related phosphene size to stimulation current, whereas Figure 3D shows the saturation effect when current _spread_ is varied (i.e., standard deviation of the Gaussian). How is stimulation current and current spread related and is there any way to increase stimulation current in your simulations?
5. In Figure 3C, why was the range of eccentricities for the Utah array simulation limited to ~4-6 degrees for V1 and ~3-4 degrees for V2? To generate the phosphene sizes for other eccentricities, could you "move" the Utah array "across cortex" and simulate parts of cortex that would have receptive fields at higher or lower eccentricities?
6. If you performed the same simulations on a randomly initialized model (or any other model that is less good at neural response predictions), what would the resulting phosphenes look like? By doing this, I hope to understand how strong of a benchmark this "phosphene-geometry analyses" is for evaluating and for differentiating models of visual cortex.

### References
- Bashivan, P., Kar, K., & DiCarlo, J. J. (2019). Neural population control via deep image synthesis. Science, 364(6439), eaav9436.
- Bosking, W. H., Sun, P., Ozker, M., Pei, X., Foster, B. L., Beauchamp, M. S., & Yoshor, D. (2017). Saturation in phosphene size with increasing current levels delivered to human visual cortex. Journal of Neuroscience, 37(30), 7188-7197.
- Cadena, S. A., Denfield, G. H., Walker, E. Y., Gatys, L. A., Tolias, A. S., Bethge, M., & Ecker, A. S. (2019). Deep convolutional models improve predictions of macaque V1 responses to natural images. PLoS computational biology, 15(4), e1006897.
- Olah, C., Mordvintsev, A., & Schubert, L. (2017). Feature visualization. Distill, 2(11), e7.

---

### Official Review · Reviewer_UMee · 2022-10-15
**Review of "Adapting Brain-Like Neural Networks for Modeling Cortical Visual Prostheses"**

**Rating:** 7
**Confidence:** 3

**Review:**

This paper presents a proof of concept model that attempts to simulate visual percepts created by a particular pattern of cortical stimulation. From a given patten of electrode activation, the authors simulate cortical activation, from which they estimate activation in portions of the visual field, from which they reconstruct the predicted percept. The authors use a combination of novel techniques and established mappings (e.g. between visual field regions and visual cortex) to achieve this. The overall model is clever and interesting, and represents a promising proof-of-concept for inferring visual percept given an activation pattern on an electrode array.

Pros:
- This paper seems important. Being able to simulate this is a key step in designing better electrode systems. Designing and refining this model seems like it would be a net good for society.
- This is a complex model, with many choices along the way. Overall, I found that the choices the authors made in their mode design were well-reasoned (although I would like to see some discussion of how robust their results are to these choices - do different choices lead to vastly different results).
- The initial results reported here are promising, and suggest that there is validity to this approach

Cons
- At this point, the way that the model is evaluated is quite qualitative. The paper would feel much stronger if there were more comparisons, or more quantitative comparisons.
- I am not convinced that the fact that the model predicts white-ish phosphenes is strong evidence that the model matches the actual percepts. Based on how the phosphenes are simulated (as the weighted average of all stimuli for that location), it seems like they would have no other option but to be white. Was it possible for phosphenes to be created, using this method of averaging all stimuli patches, what would NOT be white? Based on my understanding, I think that since this is not a property that could meaningfully vary under different experimental conditions, it cannot be taken to arbitrate on the validity of the method
- The authors claim that their model reproduces trends in phosphene size, but it appears that it only predicts a relationship between eccentricity and phosphene size for one of the two modeled systems. It is unclear if both systems should show this relationship (in which case their claims of reproducing the subjective experience of both systems is too strong), or if only one system shows it (in case the model’s failure to show it for the other system is an even stronger argument for the validity of the model)

There are some distinct weaknesses in the project at this stage. However, in spite of these weaknesses, I recommend that the paper be included in the workshop. The authors have framed this as a “proof-of-concept”, and thus by definition quite preliminary. I believe that the premise is important, and that the work could be improved by discussion with other scientists in the workshop venue. If the workshop’s primary purpose is to share novel science and get feedback to improve it, I feel that this paper is a good fit.